# Eliciting Herders' Willingness to Accept Grassland Conservation: A Choice Experiment Design in Pastoral Regions of China

**Xinxin Lv [1], Mingxue Zhang [1] and Dongqing Li [2],***

[1]  China Center for Agricultural Policy, School of Advanced Agricultural Sciences, Peking University, Beijing 100871, China

[2]  Institute of Agricultural Economics and Development, Chinese Academy of Agricultural Sciences, Beijing 100081, China

*  Correspondence: dongqing202@caas.cn

**Abstract:** Top-down grassland conservation policies are widely used to protect grassland ecosystems from degradation in developing counties. However, an inability to meet local herders' preferences when implementing such ecological policies may weaken their outcomes. Using a choice experiment design, this paper evaluated herders' willingness to accept (WTA) different possible implementations of a grazing ban policy, which is an ongoing but inflexible grassland protection policy in China. The results showed that herders were more likely to accept a grazing ban policy that targets private benefits rather than public benefits. In particular, herder's WTA decreased when the policy objective changed from improving private grassland productivity to protecting grassland wildlife (or preventing sandstorms). Additionally, broader coverage and a longer duration also increased herders' WTA a grazing ban policy; i.e., herders preferred a grazing ban policy with less coverage and a shorter duration. Our heterogeneity analysis showed that herder's WTA is not only associated with their socioeconomic characteristics, but also with their altruism. Herders with higher altruistic tendencies were more willing to engage in a gazing ban policy targeting public benefits. These findings offer valuable insight into potential methods of redesigning top-down grassland protection policies and incentivizing small herders to adapt to environmentally friendly practices in China or other countries with similar backgrounds.

**Keywords:** grazing ban policy; choice experiment; willingness to accept; altruism

## 1. Introduction

Top-down grassland conservation policies have been widely used to protect grassland from degradation in China. Grassland, accounting for over 40% of the nation's total land area in China [1], plays an essential role in the provision of essential ecosystem functions, such as the protection of biodiversity, water supply and regulation, carbon storage and climate mitigation, pollination, and cultural services [2,3]. However, human activities, such as overgrazing, and climate change have reduced many natural grasslands to poor conditions [4–7]. To alleviate the negative interference of human practices in grassland conservation, a national top-down payment-for-ecosystem-services (PES) program, the Grassland Ecological Compensation Policy (GECP), was implemented in 2011 in the main pastoral regions of China.

However, existing studies have shown that the impact of this top-down GECP program on grassland quality is limited [8], mainly because the unique payment method of the GECP fails to meet herders' heterogeneous willingness-to-accept (WTA) criteria. As the primary land users and ecosystem service providers, herders' perceptions and grazing practices largely determine the effectiveness of the ecological policy. Some herders may rank the grassland quality first, while others may rank livelihood security first [9]. Thus, eliciting

herders' WTA and fostering their positive attitudes towards flexible ecological conservation policies is essential for protecting grassland [10–12]. However, few existing studies have targeted herders' WTA grassland conservation policies. The lack of clear knowledge about herders' WTA makes it difficult to determine how we can improve ecological conservation in practice.

To elicit ecosystem providers' preferences and estimate the economic value of the attributes of ecological conservation policies, we employed a widely used stated preference-based method, i.e., choice experiments (CEs) [13]. Although numerous studies have focused on identifying ecosystem providers' WTA with respect to delivering environmental services using the CE method, very few have focused on grassland ecosystems and the attributes of policy objectives [14–17]. For example, using a CE design, Tyrväinen, Mäntymaa [15] identified forest owners' average WTA a pay-for-ecosystem-service (PES) initiative, which ranged from EUR 191 to EUR 795 per hectare per year. Common goals of a grassland ecological conservation programs, such as the GECP, are to increase herders' income and to protect grassland. However, this policy also has positive external impacts, such as the protection of biodiversity and sandstorm prevention. The respondents may therefore show different WTA the accomplishment of different objectives. Moreover, the policy imposes constraints on herders' grazing activities, which may conflict with their traditional perception of grazing as a basic right [18].

In this paper, we explored herders' WTA and the heterogeneity of grazing ban policies by conducting a CE involving herders in two main pastoral provinces in China. In our experiment, the respondents were asked face-to-face to choose from among a choice set of grazing ban policy designs, and thereby make trade-offs between policy objectives, coverage, duration, and subsidy provided by grassland rentals. We also identified the level of heterogeneity with respect to different herders' characteristics, especially focusing on altruism, as indicated by donation to charities [19,20], and its impact on herders' WTA. The main results indicated that herders prefer participating in a grazing ban policy that targets individual benefits (i.e., improving soil nutrients and forage quality on individual areas of grassland) rather than public benefits (i.e., protecting grassland wildlife in the local province or wind prevention and sand fixation in other provinces). Meanwhile, the herders' WTA increased as the policy coverage or duration increased. Another enlightening finding is that the herders with higher altruistic tendencies were more willing to accept a gazing ban policy with public benefits.

This study contributes to the literature in three aspects. Firstly, most studies on providers' WTA measures for the protection of local ecosystems have typically focused on forests or farmland [14–17], and few have used CEs to investigate herders' WTA in terms of the optimum compensation levels. By using a CE approach, our experimental design of grazing ban policies allowed us to identify the herders' WTA measures for grassland conservation. Secondly, this study explored the heterogeneous impact of the herders' degree of altruism on their WTA, which enriches the existing literature that focuses on the impacts of individual socioeconomic statuses on WTA [21,22]. Environmentally friendly choices have positive external effects and may be preferred by individuals with higher levels of altruism [23,24]. This study indicated that herders' altruism is correlated with their WTA in terms of PES programs that target ecological achievements, such as maintaining the biodiversity of a grassland landscape. Thirdly, this study used local grassland rentals to provide the payment of the herders, in exchange for their choice, for the first time, which provides a market-based solution to the compensation standard in PES programs. These results offer valuable insight into the effective leveraging of small herders' participation in grassland PES programs in China and other countries with similar backgrounds.

The remainder of this paper is organized as follows: Section 2 outlines the background of the GECP policy; Section 3 describes the CE experimental design and data collection; Section 4 presents the empirical model specification; and Section 5 provides the empirical results. Finally, Section 6 summarizes the paper and discusses the policy implications.

## 2. Background

The Gansu and Qinghai provinces, our field survey area, are the two major pastoral provinces of China, with grassland accounting for more than 20% of the total land area [1]. This grassland ecosystem supports the livelihoods of nearly two million herders with grazing livestock as their most important source of income [25]. Besides livestock production, grassland plays a dual role in the construction of an ecological civilization. Grassland ecosystems are global melting pots of biodiversity in Gansu and Qinghai, which are home to many endangered species, such as the Tibetan antelope, snow leopard, Bactrian camel, and wild yak [26]. There are 312 nature reserves in Gansu and Qinghai [27], which cover approximately 36% of the area of these two provinces. The grassland ecosystems also play an important role in preventing sandstorms, thus providing ecological benefits to the people inside and outside these regions. According to Ouyang, Song [26], the monetary value of sandstorm prevention is 31.7 billion yuan in Qinghai, which is approximately 17.1% of the total value of ecosystem functions.

However, grassland has been subject to continuous degradation caused by overgrazing, population rises, and climate change, resulting in serious economic and ecological losses of ecosystem functions [28–30]. For example, due to grassland degradation, the economic loss is approximately $20,000 per hectare per year in the severely degraded Qinghai–Tibetan Plateau [31]. In order to improve the grassland quality and increase herders' incomes, China implemented the GECP in 2011 in pastoral and semi-pastoral provinces, in which Gansu and Qinghai are included. This subsidy project was designed to be implemented in five-year periods: GECP-I (2011–2015), GECP-II (2016–2020), and the current GECP-III (2021–2025). Over 170 billion yuan has been invested in the GECP, and, by 2020, 12 million herders had been paid for their participation. Even though it is the largest PES grassland conservation program in the world in terms of the coverage area, number of participants, and the total monetary transfers, the GECP design must still be improved in order to enhance its effectiveness, especially in terms of grazing ban policies.

Grazing ban policies, one of the important measures of the GECP, have faced the most doubts and challenges from herders during the implementation of the GECP. For the conservation of grassland ecosystems, the grazing ban policy imposes total grazing prohibitions, either annually or over a certain period (e.g., spring) of a year, in severely degraded areas or ecological reserves. As herders cannot graze their livestock during the grazing prohibition periods, the government pays compensation to households according to the area covered by the grazing ban policies, with a fixed amount per unit area, usually a criterion established for a county. The fixed compensation per unit area may not cover every herder's costs, as some herders can boost their productivity with better grazing management. As a result, the public welfare benefits of grassland ecosystems, such as the conservation of biodiversity and prevention of sandstorms, may be greater than the compensation during the implementation of a grazing ban policy. Although they are protectors, local herders cannot achieve the total public gain. If the compensation is not based on the local conditions of the market or does not meet the WTA criteria of local herders, a grazing ban policy may not be welcomed. As grazing is the main activity by which herders gain an income, and government statistics show that 62.5% of the income of herders in the pastoral regions in Gansu and Qinghai originates from animal husbandry [25], it is essential for local governments to choose a suitable design for grazing ban policies.

## 3. Experimental Design and Data

### 3.1. Choice Experiment Design

Based on a pre-test survey of the field and official documents of the GECP, we selected four attributes to construct grazing ban policies, including the objectives, coverage, duration, and subsidy payment, for the choice experiment. The attributes and corresponding levels are shown in Table 1. The "objective" attribute refers to the goals of a grazing ban policy. Although grassland provides many important ecosystem functions, such as the provision of habitats and control of erosion, herders may care more about improving the

productivity of their pasture rather than other ecosystem functions that benefit the public. Different ecosystem functions may act as substitutes or complements. For example, if herders provide more livestock products using their grassland, the grassland may have less capacity to perform other ecosystem functions. Likewise, if the grassland is conserved for the provision of regulation services, herders cannot use it for livestock production. We therefore aimed to explore how policy objectives affect herders' willingness to accept policy implementation. We chose three levels for the grazing ban objective: first, to improve the soil nutrient and grassland productivity of privately managed grassland; second, to protect grassland wildlife in the local province; and third, to prevent sandstorms in other provinces. We expected that herders would care more about their grassland productivity than protecting grassland wildlife and preventing sandstorms.

**Table 1.** Attributes and levels of the choice experiment.

| Attributes | Levels |
|---|---|
| Objective | Improving soil nutrient and grassland productivity of privately managed grassland, protecting grassland wildlife in the local province, preventing sandstorms in other provinces |
| Coverage | 20%, 50%, 80%, 100% |
| Duration | One year, three years, five years |
| Subsidy (percentage of the average land rental price in the village) | 50%, 100%, 200%, 300% |

"Coverage," the percentage of grassland where grazing was banned for conservation purposes, is another important factor in herders' decision-making processes. One aspect to note is that a grazing ban policy requires the grassland be void of production for a whole year. A greater grassland coverage means that herders have less grassland left for grazing. The possible benefits and associated risks of this policy are the two major factors dictating herders' decisions regarding the proportion of grazing versus the amount of banned grassland. We selected four levels for this attribute, including 20%, 50%, 80%, and 100%.

"Duration," the length of time that a policy lasts for, also affects herders' willingness to accept a ban policy. The government generally prefers a relatively long policy, as the restoration of grassland ecosystems is a long-term process. However, herders may prefer a relatively short implementation length, as they face uncertainties in the long run. For example, herders may expect a higher return from grazing grassland as livestock prices increase in the near future. They would lose out if the policy were to subsidize them at a constant price. Adopting a ban policy also entails less flexibility in the management of grassland. According to the pre-test survey, most grassland rental contracts are only one year long, and the price is renegotiated the following year. As a grassland rental contract generally lasts for one year, and the duration of the current grazing ban policy is five years, we therefore selected one, three, and five years as the levels for the "duration."

"Subsidy" refers to the cost attribute included in the choice experiment regarding a grazing ban policy. This cost attribute represents the payment with which the government compensates herders for their conservation behavior, paid in yuan per hectare per year. It reflects herders' willingness to participate in a ban policy. When all other attributes are identical, a higher payment increases the likelihood of participation. As the grassland rental price is the opportunity cost of a grassland grazing ban policy, we used this price as the benchmark for the policy subsidies. As the grassland quality and corresponding grassland rental prices vary greatly across different villages, we used the proportion of the village land rental price as the payment amount of the policy. We chose 50%, 100%, 200%, and 300% of the land rental price (yuan/ha/year) in the village where the herders live as

the "subsidy" levels. The herders were informed of the average grassland rental price at the village level at the beginning of the choice experiment.

After selecting the attributes and their corresponding levels, we created choice sets using an orthogonal design method. Because of the incentive compatibility of a two-alternative choice or referendum [32], we included three alternatives in one choice set, including two hypothetical grazing ban policies and a "neither" alternative. A full factorial design would have generated the most comprehensive choice sets, including 144 alternatives and over 10,000 choice sets. However, it would have been too costly to conduct all of these. We therefore applied an orthogonal design and obtained 21 representative choice sets using Sawtooth software. The alternatives and choice sets were designed at the same time using the orthogonal design, so that the attribute levels were orthogonal within and across the alternatives. The orthogonal design is popular not only because it can reduce the design size and represent the results of a full factorial design, but also because it eliminates correlations between attributes [13]. To avoid respondent fatigue (i.e., each respondent taking on too many tasks [33]), we divided these 21 choice sets into three versions, with each version including seven different choice sets. One of the three versions was randomly assigned to each herder. Table 2 shows an example of a choice set.

**Table 2.** An example of a choice set.

| Attributes | Option A | Option B | Option C |
|------------|----------|----------|----------|
| Objective | Improving soil nutrient and grassland productivity of privately managed grassland | Protecting grassland wildlife in the local province | Neither |
| Coverage | 20% | 50% | |
| Duration | One year | Five years | |
| Subsidy payment | 50% of the average land rental price in the village | 100% of the average land rental price in the village | |
| Your choice | ○ | ○ | ○ |

### 3.2. Sampling Method and Data Collection

To implement the choice experiments, we conducted a field survey of the herdsmen in the Gansu and Qinghai provinces. We used a stratified random sampling strategy to select our baseline sample in 2017. The major grassland type in Gansu is alpine meadow. In Gansu, therefore, all counties were divided into four quantiles according to the income per capita of the rural residents, and one county in each quantile was randomly selected. In Qinghai, the counties were divided into three terciles according to three specified major grassland types. We then divided all the counties in each tercile into two groups based on the income per capita of the rural residents, and one county in each group was randomly selected. Then, the townships in each selected county were classified into three groups, and one township was randomly selected from each group. Two villages were randomly selected from each sampled township, and six households were randomly interviewed from each sampled village. As a result, 10 counties, 30 townships, 60 villages, and 360 households were included in our study sample. The sample is shown in Table A1 in Appendix A. In 2020, we conducted a new round of surveys to track these samples. After discounting 4 missing observations, we had 356 herders with full information.

The questionnaire included three parts, including the choice experiment, basic socioeconomic questions, and the altruism experiment. We conducted a face-to-face interview with each herder. The choice experiment of the herders was conducted by trained enumerators following a standardized procedure. The respondents were told to choose their most preferred policies from two hypothetical policies and the "neither" option in each choice set. In order to ensure the standardization and consistency of the experiment, each enumerator held a card that included an explanation of the grazing ban policy, the policy attribute description, the village-level grassland rental price, and the choice set. Before the choice experiment, the enumerator read the words on the card to the herders. Because

some herders cannot read Mandarin, the choice sets were presented in both Mandarin and Tibetan. During the experiment, the enumerator presented each alternative on the card in a standardized manner, without adding any other words or personal preferences.

In addition to the choice experiment, we also asked the herders some detailed questions in order to obtain basic individual and household information. The range of characteristics used in this research are summarized in Table A2 in Appendix A, with brief descriptions for each variable. Approximately 88% of the respondents were male, because family production and management activities are generally male dominated in pastoral areas. The respondents ranged from 15 to 78 years in age, with an average age of 48 years. Among the respondents, around 81% were Tibetan and 68% were from pastoral households. On average, the respondents had received three years of education, and 21% of them were able to write in Mandarin. The average household had 4.66 family members, and only 0.91 of laborers engaged in off-farm work. The grassland area in operation was 346.30 ha on average, and the minimum and maximum were 0 and 12,667 ha, respectively. Approximately 25% of the households either leased or rented out their grassland. When we asked the herders to classify the grassland quality of their village into one of five groups (i.e., very poor, poor, general, good, and very good), they tended to classify the grassland quality into the good group. The annual household gross income was 43,100 yuan and the value of living was 143,800 yuan on average.

Following Carpenter and Myers [19], we designed a payment-consequential donation experiment to elicit the herders' altruism level. We asked how much the herders would denote to a specific charity organization. We selected four different types of organizations, including the China Green Foundation, China Children and Teenagers' Fund, China Foundation for Poverty Alleviation, and Waterdrop Medical Crowdfunding, to elicit their maximum donation. It should be noted that the herders received a certain endowment (equal to approximately half of their daily labor wage) if they agreed to be interviewed. After they answered the four donation questions, we randomly selected one charity, and the herders donated the stated amount of money. That is, they authorized us to donate the stated amount of money to the specific charity organization. Finally, we used the maximum donation percentage among the four types of organizations, i.e., the maximum amount of the donations among the four charity organizations divided by the endowment amount, to indicate the herders' levels of altruism. The herders were further divided into two groups: the high and low altruism groups.

Figure 1 describes the herders' altruism levels. For more detailed information on the herders' altruism levels, the reader should refer to Table A3 in Appendix A. The herders paid the most to Waterdrop Medical Crowdfunding (9.3 yuan per household, equal to 20% of their endowment), followed by the China Children and Teenager's Fund (8.24 yuan, or 18%), China Green Foundation (6.97 yuan, 15%), and China Foundation for Poverty Alleviation (6.6 yuan, 14%). The maximum donation amount was 13.1 yuan per household, accounting for 27% of the herders' endowment.

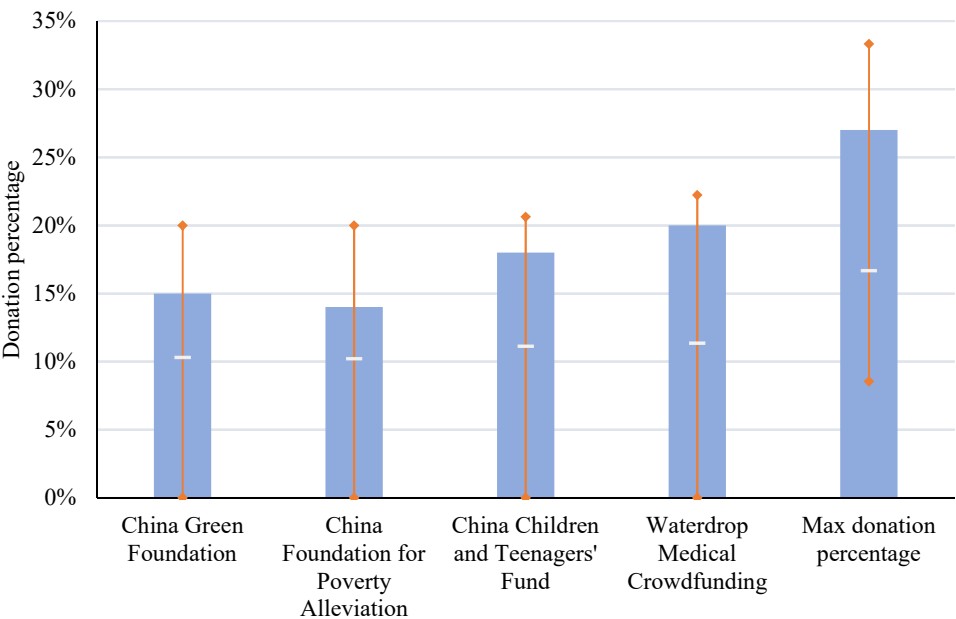

**Figure 1.** The percentage of money that the herders donated to different charity groups. The blue bar is the average donation percentage, and the white line is the median. The lower and upper ends of the orange vertical line are the 25 and 75 percent quantiles, respectively.

## 4. Econometric Model

Under the random utility framework, the utility obtained by individual $i$ from alternative $j$, $U_{ij}$, is composed of the econometric measurable component $V_{ij}$ and random component $\varepsilon_{ij}$. The $\varepsilon_{ij}$ component is unobservable and assumes an independent and identical distribution. The measurable component $V_{ij}$ depends on the attributes of the grazing ban policy, denoted by $X_j$, and individual $i$'s socioeconomic characteristics, denoted by $S_i$. Specifically, the utility of selecting alternative $j$ for individual $I$ is:

$$U_{ij} = U(X_j, S_i) = V(X_j, S_i) + \varepsilon_{ij} \tag{1}$$

The vector $X_j$ includes the objective of grazing ban $X_j^o$, the coverage $X_j^c$, the duration $X_j^d$, and the policy payment $X_j^s$. We included three alternatives in our choice set, i.e., $j \in \{A, B, C\}$. If the individual chooses alternative $j \in \{A, B, C\} \equiv J$, the probability of individual $i$ choosing option $j$, $U_{ij}$, compared to all other options $U_{ik}$ ($k \neq j$), is estimated as follows:

$$P_i(j) = Pr(U_{ij} > U_{ik}, k \neq j, k \in J) = Pr(V_{ij} + \varepsilon_{ij} > V_{ik} + \varepsilon_{ik}, k \neq j, k \in J) \tag{2}$$

$$= Pr(\varepsilon_{ij} - \varepsilon_{ik} > V_{ik} - V_{ij}, k \neq j, k \in J) \tag{3}$$

where $Pr(\cdot)$ is the probability operator. According to McFadden [34], assuming that error term $\varepsilon_{ij}$ obeys an independent and identically distributed (*i.i.d*) type-I extreme value distribution, the probability can be simplified as follows:

$$P_i(j) = e^{V_{ij}} / \sum_{n \in J} e^{V_{in}} \tag{4}$$

Assuming that the utility function is separately additive, the linear model specification is as follows:

$$V_{ij} = \beta_o X_j^o + \beta_c X_j^c + \beta_d X_j^d + \beta_s X_j^s + \beta_{asc} ASC_j + \varepsilon_{ij} \tag{5}$$

In Equation (5), $\beta_o$ is expected to be negative, as the objective of the grazing ban policy changes from improving the soil nutrient and forage quality of privately managed grassland to protecting grassland wildlife in the local province or preventing sandstorms

in other provinces, meaning that the herders are less likely to participate in the program. When the coverage and duration of the grazing ban increase, the flexibility of the land adjustment is lower, which results in the herders being less likely to participate in the grazing ban policy. We therefore expected $\beta_c$ and $\beta_d$ to be negative. The coefficient, $\beta_s$, associated with the policy subsidy, was expected to be positive, meaning that an increase in the subsidy increases the herders' utility, given that the other attributes remain unchanged. The alternative specific constant (ASC), $ASC_j$, equals 1 when the alternative is "neither" or otherwise 0. The positive coefficient $\beta_{asc}$ reflects individual *i's* tendency to refuse a grazing ban policy. We used conditional logit models to estimate the coefficients.

In order to study the heterogeneity of the herders' preferences in regard to the grazing ban policies in terms of different socioeconomic characteristics, we added the interaction term of the ASC and demographic variables to the model. The updated model was:

$$V_{ij} = \beta_s X_j^s + \beta_o X_j^o + \beta_c X_j^c + \beta_d X_j^d + \beta_{asc} ASC_j + \beta_x ASC_j \times S_i + \varepsilon_{ij} \tag{6}$$

In Equation (6), the positive coefficient $\beta_x$ can be explained as the tendency of individual *i* with demographic $S_i$ to refuse a grazing ban policy. In order to explore the influence of altruistic tendencies on the herders' marginal willingness to accept (mWTA), we added the interaction term of altruistic tendency $Altruism_i$ and $X_j^o$ to Equation (6):

$$V_{ij} = \beta_s X_j^s + \beta_o X_j^o \times Altruism_i + \beta_c X_j^c + \beta_d X_j^d + \beta_{asc} ASC_j + \beta_x ASC_j \times S_i + \varepsilon_{ij} \tag{7}$$

where $Altruism_i$ equals 1 if the herders have a high level of altruism, or otherwise 0.

In a linear and separately additive utility function, the marginal willingness to accept an attribute can be calculated by:

$$mWTA = -\frac{\partial V}{\partial X} \bigg/ \frac{\partial V}{\partial X^s} = -\beta_X / \beta_s \tag{8}$$

where $\beta_X \in \{\beta_o, \beta_c, \beta_d \}$.

## 5. Empirical Results

### 5.1. Basic Results

Firstly, we report our main results from the conditional logit models in Table 3. Column (1) represents the coefficients estimated from Equation (5). A negative coefficient means that the herders are less likely to participate in a ban policy as the attribute transitions from the baseline to a specific level, while a positive coefficient indicates that they are more willing to participate. As the estimated coefficients cannot be interpreted directly, we calculated the marginal effects and presented them in column (2). The marginal effect can be interpreted as the change in the likelihood that the herders will accept a ban policy when the attribute transitions from the baseline to a specific state. Finally, we calculated the marginal willingness to accept (mWTA) the policy using Equation (8), measured using the percentage of the village-level grassland rental price. We then transformed the percentages into absolute values by multiplying them by the village-level rental price, and we presented the mWTA in Figure 2.

The herders cared more about private policy objectives than protecting grassland wildlife and preventing sandstorms. The coefficients of the wildlife protection and sandstorm prevention variables were negative at the 1% significance level, indicating that the herders were less likely to accept a grazing ban policy when the objective was to protect grassland wildlife in the local province or prevent sandstorms in other provinces compared to a policy aiming to improve the soil nutrient and grassland productivity of their own privately managed grassland. When the objective changed from improving soil nutrient and grassland productivity to wildlife protection and sandstorm prevention, the likelihood that the herders would participate in a policy decreased by 5.6% and 7.2%, respectively. In terms of the mWTA, when the objective changed from improving soil nutrient and

grassland productivity to wildlife protection (sandstorm prevention), the herders' mWTA increased by 291.6 and 370.4 yuan/ha/year, respectively.

**Table 3.** The basic results of the herders' WTA.

| Variables | Coefficients (1) | Margins (2) |
|---|---|---|
| Baseline: the level of objective is improving herders' own grassland quality | | |
| Wildlife protection | −0.236 *** | −0.056 *** |
| | (0.074) | (0.018) |
| Sandstorm prevention | −0.300 *** | −0.072 *** |
| | (0.074) | (0.018) |
| Baseline: the level of coverage is 20% | | |
| Coverage 50% | −0.399 *** | −0.095 *** |
| | (0.092) | (0.022) |
| Coverage 80% | −0.534 *** | −0.127 *** |
| | (0.093) | (0.022) |
| Coverage 100% | −0.684 *** | −0.163 *** |
| | (0.090) | (0.021) |
| Baseline: the level of duration is 1 year | | |
| Duration 3 years | −0.139 * | −0.033 * |
| | (0.075) | (0.018) |
| Duration 5 years | −0.195 ** | −0.046 ** |
| | (0.077) | (0.018) |
| ASC | 0.420 *** | 0.100 *** |
| | (0.111) | (0.026) |
| Land rent multiple | 0.358 *** | 0.085 *** |
| | (0.036) | (0.008) |
| Observations | 7497 | 7497 |
| LR chi2 | 344.72 | — |

Note: this table shows the basic results from the conditional logit models. Column (1) represents the coefficients estimated using Equation (5). Column (2) represents the marginal effects, interpreted as the change in the likelihood that herders will accept the policy when the attribute changes from the baseline to a specific state. Standard errors in parentheses. * $p < 0.10$, ** $p < 0.05$, *** $p < 0.01$.

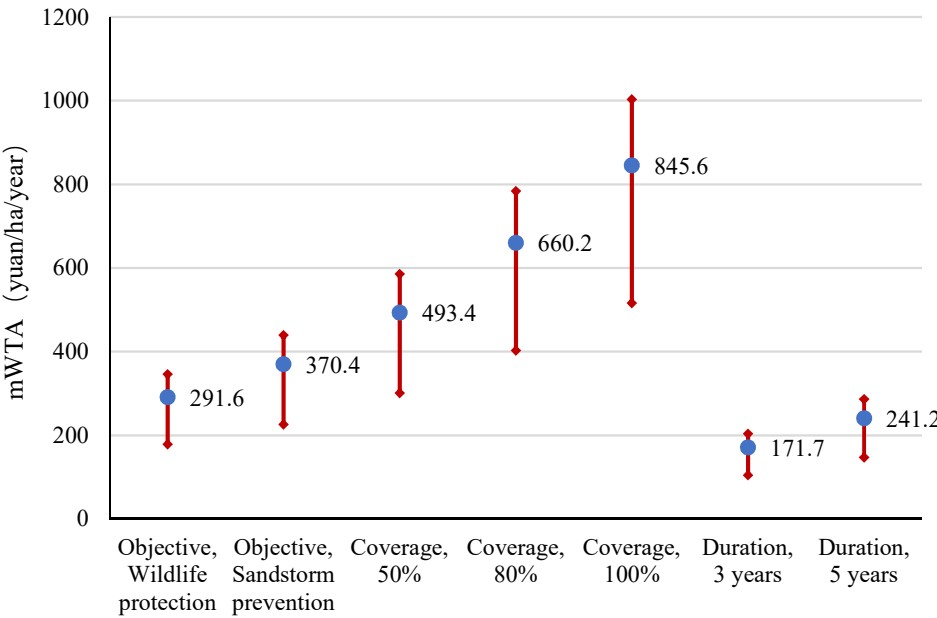

**Figure 2.** The mWTA for different attribute levels. The blue point is the average mWTA. The lower and upper ends of the red vertical line are the 25 and 75 percent quantiles, respectively.

The herders' willingness to accept a policy was higher when the policy coverage area accounted for more of their farm size. The coefficients of the policy coverage were all

negative at the 1% significance level. This means that, as the coverage area increased, the herders were more reluctant to accept a grazing ban policy. The marginal effects show that the likelihood of the herders accepting a ban policy decreased by 9.5% as the coverage increased from 20% to 50%. When the coverage increased from 20% to 80% and 100%, the likelihood decreased by 12.7% and 16.3%, respectively. According to the mWTA, when the coverage level was changed from 20% to 50%, 80%, and 100%, the mWTA increased by 493.4, 660.2, and 845.6 yuan/ha/year, respectively.

The herders' willingness to accept also increased as the length of the policy increased. Compared to a one-year policy, the coefficients of the three- and five-year policies were both negative and significant at the 10% significance level, indicating a preference for a ban policy with a shorter duration. When the duration of the ban policy changed from one year to three and five years, the herders' willingness to participate in the policy decreased by 3.3% and 4.6%, respectively. In addition, the mWTA was 171.7 yuan/ha/year for a change from one to three years, and 241.2 yuan/ha/year for a change from one year to five years.

Finally, the positive and significant index of the ASC parameter suggests that the respondents were inclined to reject a grazing ban policy. The marginal effect indicates that the possibility of the herders rejecting a grazing ban policy was 10% higher than that of the herders accepting it. This is consistent with the findings of Ho [35] and Gubo and Xinping [36], who found that most of their surveyed herders disapproved of a grazing ban policy, feeling that a ban is not suitable for conservation, or complaining about adverse income effects. Some herders even admitted to illegal (nighttime) grazing.

### 5.2. Heterogeneity Effects

The results from the heterogeneity analysis show that the herders' WTA differed among the groups with different socioeconomic characteristics (Table 4). Firstly, herders who were male, Tibetan, and could write in Mandarin were more willing to accept a grazing ban policy. In pastoral areas of China, males are the main laborers with the responsibility for grazing grassland and other family production activities. They may care more about the sustainable use of grassland. Tibetan respondents were more willing to accept a grazing ban policy, because their lifestyles are more traditional and conversative due to religion or other informal institutions. We also found that the respondents who were registered as pastoralists in the household registration system were more willing to accept a ban policy than those who were registered as non-pastoralists (urban or crop area). The herders registered as pastoralists may live in grassland and, therefore, be more concerned about the sustainable development of grassland. Similarly, those who were able to write in Mandarin were more willing to accept a ban policy, probably because they were able to gain more knowledge about the importance and benefits of grazing prohibitions.

Secondly, households that were wealthier, smaller in size, and performed more off-farm labor work were more willing to accept a ban policy. Households with more occupants were less likely to accept a grazing ban policy, because these households may suffer greater economic pressures and need to earn more money through the use of grassland as opposed to a grazing ban. Poor herders depend on grassland for their sustenance, while wealthy herders possess money to support their families and are less reliant on grassland. Therefore, wealthy herders were more willing to accept a ban policy. Households whose members performed more off-farm labor work were also more willing to accept a ban policy, because these households often have other sources of income in addition to husbandry, engage less in husbandry, and are less dependent on grassland.

Finally, those herders who participated in the grassland market, whose grassland quality was lower than the village average, and who operated on smaller grassland areas were more willing to accept a ban policy. Households that had joined the grassland rental market, which possibly had idle grassland or access to exterior channels and funds to obtain grassland for grazing, were more willing to accept a ban policy than others. Those herders who believe that grassland in the village is of a worse quality may consider it necessary to implement a grazing ban; thus, they appeared to be more willing to accept this

policy. The herders operating on larger grassland areas were less likely to participate in a ban policy, because these herders are more dependent on grassland and may lose essential sources of income by participating in the policy.

**Table 4.** Heterogenous factors of different household or individual characteristics.

| Variables | Coefficients (1) | Margins (2) |
|---|---|---|
| ASC | 0.298 | 0.069 |
|  | (0.412) | (0.096) |
| Interactions of ASC with personal characteristics |  |  |
| Gender (1 = male; 0 = female) | −0.530 *** | −0.123 *** |
|  | (0.138) | (0.032) |
| Ethnic (1 = Tibetan; 0 = others) | −0.518 *** | −0.120 *** |
|  | (0.155) | (0.036) |
| Residence (1 = herder; 0 = others) | −0.308 *** | −0.072 *** |
|  | (0.100) | (0.023) |
| Mandarin proficiency (1 = written; 0 = others) | −0.550 *** | −0.128 *** |
|  | (0.204) | (0.047) |
| Interactions of ASC with family characteristics |  |  |
| Household size | 0.159 *** | 0.037 *** |
|  | (0.024) | (0.006) |
| House value (thousand yuan) | −0.009 *** | −0.002 *** |
|  | (0.002) | (0.001) |
| Off-farm labor number | −0.156 *** | −0.036 *** |
|  | (0.0431) | (0.010) |
| Interactions of ASC with grassland characteristics |  |  |
| Grassland market participation (1 = yes; 0 = no) | −0.520 *** | −0.121 *** |
|  | (0.108) | (0.025) |
| Grassland quality | 0.191 *** | 0.045 *** |
|  | (0.050) | (0.012) |
| Grassland size (ha) | 0.087 *** | 0.020 *** |
|  | (0.029) | (0.007) |
| Attribute levels | Yes | Yes |
| Observations | 7371 | 7371 |
| LR chi2 | 546.48 | — |

Note: this table describes the relationship of herders' WTA with different socioeconomic characteristics. The ASC interacted with personal characteristics, family characteristics, and grassland characteristics, respectively. Column (1) shows the coefficients calculated using Equation (6), and Column (2) refers to the marginal effects. Standard errors in parentheses. *** $p < 0.01$.

*5.3. Altruism and MWTA*

The heterogeneity results from the perspective of altruism show that, when the policy objective changed from private objectives (i.e., improving soil nutrient and grassland productivity) to public objectives (i.e., wildlife protection and sandstorm prevention), the herders with a high level of altruism were more willing to accept a ban policy, given that all of the other attributes of the policy remained unchanged (Table 5). Specifically, the coefficients of the wildlife protection and sandstorm prevention variables were negative at the 1% significance level, showing that those herders in the low altruism group were less likely to accept a grazing ban policy with the objectives of wildlife protection and sandstorm prevention compared to a policy aiming to improve soil nutrient and grassland productivity. Moreover, when the objective changed from improving soil nutrient and grassland productivity to wildlife protection (sandstorm prevention), the coefficient of the interaction terms between wildlife protection (sandstorm prevention) and altruism indicated that those herders with high altruism were more willing to accept a grazing ban policy in comparison to the low altruism group. The marginal effects indicate that, when the objective changed from improving soil nutrient and grassland productivity to wildlife protection or sandstorm prevention, the likelihood of herders with high altruism participating in the policy decreased by 1.6% and 3.2%, respectively. Meanwhile, the likelihood was 9.1% and 9.8% in the low altruism group, respectively. In terms of the mWTA, those herders with high

altruism achieved 80.1 yuan/ha/year (481.9 yuan/ha/year + (−401.8 yuan/ha/year)) and 170.8 yuan/ha/year (521.7 yuan/ha/year + (−350.9 yuan/ha/year)) when the objective changed from improving soil nutrient and grassland productivity to wildlife protection and sandstorm prevention, respectively. Meanwhile, the mWTA was 481.9 and 521.7 yuan/ha/year in the low altruism group, respectively.

**Table 5.** The effect of altruism on the mWTA.

| Variables | Coefficients | Margins | mWTA Multiplier | mWTA |
|---|---|---|---|---|
| | **(1)** | **(2)** | **(3)** | **(4)** |
| ASC | 0.205 | 0.048 | — | — |
| | (0.414) | −0.096 | | |
| Wildlife protection | −0.390 *** | −0.091 *** | 1.089 | 481.88 |
| | (0.101) | (0.023) | | |
| Sandstorm prevention | −0.422 *** | −0.098 *** | 1.179 | 521.71 |
| | (0.099) | (0.023) | | |
| Altruism * wildlife protection | 0.325 *** | 0.075 *** | −0.908 | −401.79 |
| | (0.124) | (0.029) | | |
| Altruism * sandstorm prevention | 0.284 ** | 0.066 ** | −0.793 | −350.90 |
| | (0.122) | (0.028) | | |
| Other attributes | Yes | Yes | Yes | Yes |
| Controls | Yes | Yes | Yes | Yes |
| Observations | 7371 | 7371 | 7371 | 7371 |
| LR chi2 | 556.68 | — | — | — |

Note: this table demonstrates the influence of the altruistic tendency of the herders on their mWTA, which is represented by the interaction term of altruistic tendency and the objectives of the grazing ban. Column (1) displays the coefficients calculated using Equation (7), and Column (2) refers to the marginal effects. Columns (3) includes the mWTA multiplier, which means the multiple of the village-level rental price. Column (4) shows the mWTA, which is calculated by multiplying Column (3) with the average village-level rental price. Standard errors in parentheses. * $p < 0.10$, ** $p < 0.05$, *** $p < 0.01$.

## 6. Discussion and Conclusions

To improve the design of the ongoing top-down grazing ban policy in the pastoral area of China, we conducted a CE design to elicit herders' WTA the different attributes (i.e., objective, coverage, and duration) of such policies. Our results show that herders are more likely to accept a grazing ban policy that targets private benefits rather than public benefits. In particular, herders' WTA decreased when the policy objective changed from improving private grassland productivity to protecting grassland wildlife (or preventing sandstorms). Additionally, herders preferred a grazing ban policy with less coverage and a shorter duration. The heterogeneity analysis showed that herders' WTA is not only associated with their socioeconomic characteristics, but also with their altruism level.

The first significant result obtained from our research is that herders are likely to receive greater compensation from a grazing ban policy aiming to preserve ecosystem functions rather than a policy aiming to improve grassland productivity. Herders can claim an additional compensation of 291.6 and 370.4 yuan/ha/year for protecting grassland wildlife and preventing sandstorms, respectively, as a condition of participation. This result indicates that when the policy's purpose is more in line with the interests of herders, they are more willing to accept the policy. Therefore, policy makers can inform herders that the quality of their grassland will improve or that their living environment will be better after the implementation of the policy in the official document.

Secondly, we found that herders' WTA is improved when the duration or the coverage of the grazing ban policy increases. The government, which aims to improve the grassland ecosystem functions, may prefer a ban policy that has a longer duration and wider coverage. However, the tradeoff is that herders' WTA for such ban policies are lower, as herders may feel uncertain about a ban policy that lasts for longer duration, such as five years.

When policy makers design such ban policies, they should balance herders' WTA and the duration and coverage of the policies.

Thirdly, our results indicate that the payment standard of the ongoing GECP policy is too low, which may weaken the effects of the policy in improving the grassland quality. With the same policy objectives, the herders' WTA was 1087 yuan/ha/year higher for a five-year policy with 100% coverage than that for a one-year policy with a 20% coverage. However, the payment standard of the ongoing grazing ban policy is only 112.5 yuan/ha/year. Because of the failure of the GECP's payment standard to meet herders' WTA, some herders tend to graze illegally in order to support their livelihoods, which makes it hard for the GECP to achieve the expected goal [37]. The literature suggests that the total monetary value of grassland ecosystem functions is 10,876 yuan/ha/year [38]. It is therefore still possible to enhance welfare by increasing the subsidy so as to encourage herders to participate in the GECP.

Finally, households that perform more off-farm labor work and participate in the grassland market were more willing to accept a ban policy. The households who participate in the off-farm labor market have fewer laborers engaging in husbandry and tend to graze fewer livestock. Therefore, they are less likely to rely on grassland. The grassland rental market helps to increase the number of off-farm jobs and release the pressure on the grassland as a source of herder livelihood. [39]. These results indicate that a well-functioning labor market and grassland rental market are helpful for the implementation of the GECP. Our results are consistent with Hu, Huang [37], and Hou, Xia [8], who showed that off-farm employment can reduce over-grazing and improve grassland quality.

Additionally, herders with high altruism are more likely to accept a ban policy. The values of the marginal WTA were 481.9 and 521.7 yuan/ha/year in the low and high altruism groups, respectively. Considering the importance of various ecological service functions provided by grassland to human beings, we can glean that herders with high altruism are prone to pay attention to the service functions of the grassland benefiting the public and show stronger willingness to protect the grassland. This result shows that cultivating the altruistic tendencies of herders could increase their participation in a ban policy.

Although this paper provides valuable results and suggestions for improving the on-going ban policy, it has at least three limitations. Firstly, the results of this paper rely on a hypothetical CE design, whose hypothetical bias could negatively affect the results. However, due to budget limitations, we were unable to conduct an incentivized choice experiment. We call for future research to use policy consequential or payment consequential CE designs to elicit more accurate measures of herders' WTA. Secondly, our data sample can only represent the Qinghai and Gansu provinces, while it lacks the ability to represent the whole northern pastoral area of China. Our heterogeneity analysis showed that herders' WTA may vary according to many socioeconomic characteristics. China's northern pastoral area covers about 90% of the total national grassland area [1], which hosts several different grassland types and people from different ethnic groups and economic levels. We therefore emphasize that our results are more suitable for interpreting the Qinghai and Gansu provinces. Finally, the analysis is limited by the conditional logit models. This is not a major problem because of the simplicity of the design and the independence of the attributes. Future analyses could apply more sophisticated approaches, such as random parameter models, to improve the accuracy of the results.

**Author Contributions:** Conceptualization, X.L.; methodology, X.L.; software, X.L.; validation, X.L., M.Z. and D.L.; formal analysis, X.L.; investigation, X.L., M.Z. and D.L.; resources, X.L.; data curation, X.L. and M.Z.; writing—original draft preparation, X.L., M.Z. and D.L.; writing—review and editing, X.L., M.Z. and D.L.; visualization, X.L. and M.Z.; supervision, D.L.; project administration, M.Z.; funding acquisition, X.L. and M.Z. All authors have read and agreed to the published version of the manuscript.

**Funding:** This research was funded by the National Natural Sciences Foundation of China (grant number 72173004, 71773003) and the Major Consulting Project of Chinese Academy of Engineering grant number (grant number 2022-HZ-09).

**Institutional Review Board Statement:** Not applicable.

**Informed Consent Statement:** Not applicable.

**Data Availability Statement:** Not applicable.

**Conflicts of Interest:** The authors declare no conflict of interest.

## Appendix A

**Table A1. Sample**.

| Province | County | Town | Village | Household |
|----------|--------|------|---------|-----------|
| Gansu | 4 | 12 | 24 | 144 |
| Qinghai | 6 | 18 | 36 | 216 |
| Total | 10 | 30 | 60 | 360 |

Note: for each province, we combined annual income per capita, grassland type, and geographical position. We sampled 3–4 townships from each county, 2–3 villages from one township, and 6–9 households, which are randomly selected from each village.

**Table A2.** Descriptive statistics of the samples.

| Variable | Describe | Mean | SD | Min | Max |
|----------|----------|------|-----|-----|-----|
| a. Personal level (herder who answered our questionnaire) | | | | | |
| Gender | 1 = male; 0 = female | 0.88 | 0.33 | 0 | 1 |
| Age | Age of interviewee | 47.99 | 11.84 | 15 | 78 |
| Ethnicity | 1 = Tibetan; 0 = other | 0.81 | 0.39 | 0 | 1 |
| Residence | 1 = herder; 0 = other | 0.68 | 0.47 | 0 | 1 |
| Education | Years in education | 3.13 | 4.23 | 0 | 16 |
| Mandarin proficiency | 1 = written; 0 = other | 0.21 | 0.41 | 0 | 1 |
| b. Household level | | | | | |
| Household size | The number of people in the household | 4.66 | 1.92 | 1 | 12 |
| Off-farm labor number | The number of labors engaged in off-farm work | 0.91 | 1.09 | 0 | 5 |
| Grassland size | Operated grassland area (ha) | 346.30 | 1302.55 | 0 | 12,667 |
| Grassland market participation | Either rent in or rent out grassland (1 = yes) | 0.25 | 0.43 | 0 | 1 |
| Grassland quality | The grassland quality of your village (1 = very poor; 2 = poor; 3 = general; 4 = good; 5 = very good) | 3.82 | 0.89 | 1 | 5 |
| Income | Annual gross income (thousand yuan) | 43.09 | 111.70 | −498 | 570 |
| House value | The value of the resident house (thousand yuan) | 143.84 | 204.78 | 0 | 1800 |

**Table A3.** Description of the herders' altruistic tendencies.

| Variable | Mean | SD | Min | Max |
|---|---|---|---|---|
| Endowment | 49.89 | 14.06 | 11 | 68.5 |
| a. The amount of money that the herder donated to different agencies | | | | |
| China Green Foundation | 6.97 | 9.83 | 0 | 68.5 |
| China Foundation for Poverty Alleviation | 6.60 | 9.23 | 0 | 68.5 |
| China Children and Teenagers' Fund | 8.24 | 10.27 | 0 | 68.5 |
| Waterdrop Medical Crowdfunding | 9.29 | 12.75 | 0 | 68.5 |
| Maximum donation amount | 13.06 | 14.57 | 0 | 68.5 |
| b. The percentage of the money that the herder donated to different agencies | | | | |
| China Green Foundation | 0.15 | 0.22 | 0 | 1 |
| China Foundation for Poverty Alleviation | 0.14 | 0.19 | 0 | 1 |
| China Children and Teenagers' Fund | 0.18 | 0.23 | 0 | 1 |
| Waterdrop Medical Crowdfunding | 0.20 | 0.26 | 0 | 1 |
| Maximum donation percentage | 0.27 | 0.29 | 0 | 1 |

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
