# Peer review of "Eliciting Herders’ Willingness to Accept Grassland Conservation: A Choice Experiment Design in Pastoral Regions of China"

_land, doi:10.3390/land11091463_

Round 1
Reviewer 1 Report
This is an interesting paper which should be taken into consideration for publication. However, I have spotted several issues in the paper which must be further revised:
1) The introduction, methods and results are nicely presented. I think that in the introduction it can be mentioned also an interesting case of pastureland degradation and poverty of herders in Mongolia (see W. Lise et al in journal Ecological Economics, 2006) and see a case of herders/shepherds' fight for their rights to grazing and their traditional identity (see an article of Thomas O'Brien et al, 2019 in journal Identities on the role of identity in shepherd protests in Romania). Also, it can be shortly mentioned in the literature review or in the discussion of the paper the role of herders' guarding dogs in their fight with carnivores (see Gozales et al, 2012 in journal Human-Wildlife Interaction) and the ethics of cullings dogs (including herder guardian dogs) - see an article on protests against dog culling in journal Area, 2015). Finally, the method section should present the limits of this method and data used as well as the broader limits of the paper.
2) There are no discussions in this paper. Authors have to write either a new section of about 3 paragraphs or to add 2-3 paragraphs of discussions at the end of section 5 (Results). Discussions must link the findings of this paper to the literature review used in this paper.
3) If the policy recommendations are good, I would better see them placed as a separate section before the conclusions. Moreover, the conclusions are too short. Conclusions should include that the aims of the paper have been solved, the international and regional (Asian) implications of this study and its novelty as well as how other scholars can further develop the outcomes of this paper.
Reviewer 2 Report
I was very much looking forward to reading this paper, as the topic is very interesting, but I stopped reading at line 100 because of all the grammatical errors (e.g. incorrect use of the definite article) and the sentences which simply don't make sense. A few examples:
Line 35-37 'As the primary land users and ecosystem service providers, herders’ grazing practices largely shape the effects of the ecological policy' - what does shape the effects of ecological policy mean?
Line 57 - 'respondents were asked to choose among a choice set of grazing ban policy design face-to-face' - ???
Line 79-80 - 'This study indicates that herders’ altruism is corrected with their WTA in PES programs that targeted to ecological achievement' - this makes no sense.
The paper needs to be revised by a native English speaker, since in its current form it is not intelligible.
Round 2
Reviewer 1 Report
Authors have done a good revision and I would like to accept the paper for publication. Just for the final author correction stage it would be good to change the title of the final section from Conclusion and discussion into Discussions and conclusions.
Author Response
We thank the reviewer for the positive evaluation of the previous manuscript. We have changed the title of the final section from Conclusion and discussion into Discussions and conclusions.
Reviewer 2 Report
The authors have addressed my comments and the result is a well-written and interesting paper.
Author Response
We thank the reviewer for the positive evaluation of the previous manuscript.